# Structure and Activity of a Cytosolic Ribosome-Inactivating Protein from Rice

**DOI:** 10.3390/toxins11060325

**Published:** 2019-06-06

**Authors:** Jeroen De Zaeytijd, Pierre Rougé, Guy Smagghe, Els J.M. Van Damme

**Affiliations:** 1Department of Biotechnology, Faculty of Bioscience Engineering, Ghent University, Coupure links 653, B-9000 Ghent, Belgium; Jeroen.DeZaeytijd@UGent.be; 2Department of Plants and Crops, Faculty of Bioscience Engineering, Ghent University, Coupure links 653, B-9000 Ghent, Belgium; Guy.Smagghe@UGent.be; 3UMR 152 PharmaDev, Université Paul Sabatier, Institut de Recherche et Développement, Faculté de Pharmacie, 35 Chemin des Maraîchers, 31062 Toulouse, France; Pierre.Rouge@free.fr

**Keywords:** rice, ribosome-inactivating protein, recombinant protein expression, localization, structure, enzymatic activity

## Abstract

Ribosome-inactivating proteins (RIPs) are cytotoxic enzymes that inhibit protein translation by depurinating ribosomal RNA. Although most plant RIPs are synthesized with leader sequences that sequester them away from the host ribosomes, several RIPs from cereals lack these signal peptides and therefore probably reside in the cytosol near the plant ribosomes. More than 30 RIP genes have been identified in the rice (*Oryza sativa* spp. japonica) genome, many of them lacking a signal peptide. This paper focuses on a presumed cytosolic type-1 RIP from rice, referred to as OsRIP1. Using 3D modeling it is shown that OsRIP1 structurally resembles other cereal RIPs and has an active site that meets the requirements for activity. Furthermore, localization studies indicate that OsRIP1-eGFP fusion proteins reside in the nucleocytoplasmic space when expressed in epidermal cells of *Nicotiana benthamiana* or *Arabidopsis thaliana* suspension cells. Finally, OsRIP1 was recombinantly produced in *Escherichia coli* and was demonstrated to possess catalytic activity. Interestingly, this recombinant RIP inactivates wheat ribosomes far less efficiently than rabbit ribosomes in an in vitro system. These findings raise some interesting questions concerning the mode of action and physiological role of OsRIP1. This is the first time a RIP from rice is investigated at protein level and is shown to possess biological activity.

## 1. Introduction

Ribosome-inactivating proteins (RIPs) are enzymes that can irreversibly inhibit protein translation by removing a specific adenine residue from the conserved sarcin/ricin loop of the 28S rRNA. It has been reported that these proteins can depurinate ribosomal RNA at multiple sites and also remove adenine residues from other substrates such as herring sperm DNA, poly(A) and tobacco mosaic virus RNA. For these reasons RIPs are considered polynucleotide:adenosine glycosidases [1,2,3].

Based on their domain architecture RIPs are classified in different subgroups. Type-1 RIPs only contain a RIP domain whereas the term type-2 RIPs refers to proteins composed of an enzymatic domain linked to a lectin domain. The less common RIPs that contain a RIP domain fused to a non-lectin domain are referred to as type-3 RIPs [4,5]. Given their ribosome-inactivating activity, RIPs are considered cytotoxic proteins. Type-2 RIPs are usually more toxic towards animal cells than type-1 RIPs because they can bind glycan structures on the surface of cells improving their cellular uptake. However type-1 RIPs are also cytotoxic when they are able to reach the cytosol [6,7]. It is accepted that RIP toxicity results from the combined effects of: (i) binding to the cell surface, (ii) uptake by the cell, (iii) ability of the RIP to reach the cytosol of the host cell, (iv) intracellular stability, and (v) successful interaction with the ribosomes. 

Because of their toxicity, RIPs are presumed to play a role in the protection of the plant against plant pathogenic bacteria, fungi, and pest insects as recently reviewed by Zhu et al. [8]. The adverse effect of RIPs on eukaryotic ribosomes also implies that plant cells producing these enzymes are possibly in danger of being affected by endogenous RIPs. Type-2 RIPs in general are several-thousand-fold less active on plant ribosomes than on mammalian ribosomes [9] while type-1 RIPs can efficiently act on plant ribosomes [10]. To circumvent the possible toxic effect of their own RIPs, plants protect themselves by sequestering RIPs away from the host ribosomes. Indeed, most classical plant type-1 and type-2 RIPs are synthesized with N-terminal signal sequences that target them to the endomembrane system [11]. A lot of RIPs are subsequently targeted to the vacuole by additional carboxy terminal targeting sequences preventing them to become active before they reach the storage organelles [12,13,14]. Other RIPs like the pokeweed antiviral protein are secreted and stored in the cell wall [15]. The toxic effect of RIPs towards host cells is hypothesized to represent a defense mechanism by which damaged plant cells are eliminated through apoptosis after release of the sequestered RIPs in the cytosol [15,16,17]. 

Interestingly, next to these classical RIPs, some RIP lineages appear to be synthesized without signal peptides and therefore are suggested to reside in the cytosolic compartment where they can come into contact with the host ribosomes [18,19,20,21]. Sequences encoding these presumed cytoplasmic RIPs are predominantly, though not exclusively, found in the genomes of cereals [22]. The different subcellular localization compared to classical RIPs raises some questions concerning their mode of action. Are cereal RIPs that reside in the cytoplasm inactive against the host ribosomes or, in contrast, do they exert their physiological role by inhibiting conspecific ribosomes? Literature suggests that both options are possible. For example, the cytoplasmic maize RIP1 and barley seed RIPs are not active on conspecific ribosomes [20,23] while the leaf form of the presumed cytoplasmic wheat RIP referred to as tritin-L is capable of depurinating wheat ribosomes [24]. To make things even more complicated, the barley RIP JIP60 was shown to be active on host ribosomes only in certain conditions and exerts its function by re-organizing the translational machinery under stress conditions [19,22,25,26].

The presumed cytosolic localization of RIPs lacking leader sequences has, to our knowledge, always been postulated on the absence of a signal peptide. However, predictions of subcellular localization solely based on the presence or absence of signal sequences is risky since unconventional protein secretion pathways have been described [27]. Proteins that lack signal peptides can therefore still be secreted and be classified as so-called leaderless secreted proteins (LSP). It was reported that over 50% of plant proteins identified in the plant secretome lack a signal peptide. However, it is unclear how many of these proteins should be termed real LSPs [27]. Because of the existence of these unconventional secretion pathways the subcellular localization of proteins should be determined experimentally.

In the rice genome both RIP genes containing a signal peptide as well as RIP genes lacking signal sequences have been identified [28], suggesting that rice plants express both cytosolic and secreted type-1 RIPs. This study investigates the structure, subcellular localization and catalytic activity of OsRIP1, a presumed cytosolic type-1 RIP from *Oryza sativa*. We show that this RIP has a three-dimensional structure that resembles other cereal RIPs and an active site cavity that suggests OsRIP1 to be a functional enzyme. Furthermore, localization studies have proven that the OsRIP1-eGFP fusion protein is indeed located in the cytosol when expressed in epidermal cells of *Nicotiana benthamiana* or *Arabidopsis thaliana* suspension cells. Finally, we demonstrate that OsRIP1, recombinantly produced in *E. coli*, inhibits translation and is far less active on cereal ribosomes compared to animal ribosomes. 

## 2. Results

### 2.1. The OsRIP1 Structure Resembles Other Cereal RIPs and Suggests OsRIP1 Is a Functional Ribosome-Inactivating Protein

The OsRIP1 protein sequence was compared to sequences of well-studied cereal RIPs in a multiple sequence alignment analysis (Figure 1). This alignment revealed that the active site residues (Y79, Y112, E167, R170, and W204) that make up the specificity pocket of RIPs are conserved in OsRIP1. OsRIP1 shows the highest degree of sequence identity to the barley bRIP1 (57.9%), followed by the barley bRIP2 (57.1%) and tritin (56.8%). The sequence identity with maize RIP1 (35.1%) and JIP60 (30.5%) is far less. According to the SignalP software no signal peptide is present in the OsRIP1 protein sequence [29].

In contrast to RIPs from non-cereal plants several cereal RIPs require unique processing and activation steps. Some famous examples are JIP60 from barley and the maize RIP1 and RIP2. These RIPs contain internal peptides that are proteolytically removed. Although the bRIP1 from barley does not have a removable peptide, it was shown to have a ‘switch region’ located at an equivalent position to the proteolytic cleavage site in the maize RIPs. This switch region can represent an “open” or a “closed” form depending on the liganded state [22,30]. Based on its protein sequence, OsRIP1 does not possess an inactivation region. Furthermore, at the location of the reported ‘switch’ region the OsRIP1 protein sequence differs from those of the barley RIPs and tritin (Figure 1). Interestingly, compared to the tritin and the barley RIPs, OsRIP1 has an extra amino acid stretch near its C-terminus.

Aside from the before mentioned inactivation regions or switch regions, there are some small but important structural differences between cereal and non-cereal RIPs. Maize RIP1 and bRIP1, the only two cereal RIPs for which the crystal structure has already been determined, seem to lack one α-helix in the N-terminal subdomain of the enzymatic domain and two β-sheets in the C-terminal subdomain. Modeling of the three-dimensional structure for OsRIP1 using the Phyre2 program [31] enabled to compare this structure to the X-ray structures for the ricin A-chain (RTA), trichosanthin (TCS), maize RIP1 (B-32), barley RIP1 (bRIP1) and the model for tritin, a RIP from wheat (Figure 2). The OsRIP1 structure resembles the fold of other cereal RIPs in the sense that the α-helix in the N-terminal subdomain as well as the two β-sheets in the C-terminal subdomain are lacking in comparison to the structure of non-cereal RIPs. Like bRIP1 and tritin, OsRIP1 is characterized by the presence of an additional C-terminal α-helix which is not present in maize RIP1. To date, the function of this extra C-terminal helix is not known. Overall OsRIP1, barley RIP1 and tritin show very similar three-dimensional conformations except for the presence of the extra amino acid stretch present in OsRIP1 near the C-terminus.

More detailed homology modeling of OsRIP1 and docking of adenosine monophosphate (AMP) the nucleotide-binding site was performed using the program YASARA. The QMEAN6 score calculated for the OsRIP1 model gave an acceptable value of −1.45. The predicted OsRIP1 structure is characterized by a typical RIP fold. Interestingly, in this model the extra amino acid stretch was predicted to be a helix like structure in contrast to the model made by the Phyre2 program, which predicted a sheet-like structure (Figure 3). 

Docking experiments showed that binding of AMP in the OsRIP1 cavity is mediated by seven H-bonds and two stacking interactions (Figure 4). The electropositive character of the molecular surface surrounding the active cavity favors the accommodation of AMP to the active site of OsRIP1. Furthermore, the positively charged surface around the active site of OsRIP1 will stabilize the interaction with the electronegatively charged 28S RNA. In addition to the electropositive character of the active site, the shape and size of the cavity are key factors which are prerequisites for the proper accommodation of AMP to the active site. 

Taken together, all these structural features favor the hypothesis that OsRIP1 is a functional ribosome-inactivating protein that structurally resembles other cereal RIPs, especially the barley bRIP1 (Figure 5).

### 2.2. OsRIP1 Locates to the Nucleocytoplasmic Space

Since the coding sequence for OsRIP1 lacks a signal peptide the RIP will be synthesized on ribosomes in the cytoplasm. To investigate the subcellular localization for OsRIP1, eGFP fusion constructs were made for the transient expression in epidermal cells of *Nicotiana benthamiana*. OsRIP1 fusion proteins with either N- or C-terminal eGFP fusion both yield a fluorescent signal in the nucleus and the cytosol (Appendix A).

Co-localization experiments were performed by staining the nuclei with 4′,6-diamidino-2-phenylindole (DAPI). As a control, infiltration experiments were also conducted with constructs expressing either free eGFP or eGFP fused to the N-terminal signal peptide from a presumed secreted rice RIP (LOC_Os03g48200.1) (Figure 6).

Both OsRIP1-eGFP and free eGFP displayed a nucleocytoplasmic expression pattern. Clear co-localization between EGFP and DAPI was observed in the nucleus. The SP-eGFP construct showed a more dotted fluorescent pattern near the cell borders, resembling secretory vesicles. For the SP-eGFP construct fluorescence was present around but not inside the nucleus as evidenced by the lack of co-localization with DAPI.

Epidermal cells of *N. benthamiana* have large vacuoles squeezing the cytosol against the cell wall which makes it difficult to distinguish the cytoplasm, plasma membrane, or apoplast [32]. Therefore, a co-localization experiment was performed using propidium iodide as a cell wall marker (Figure 7). Fluorescent intensity plots of cross sections spanning the cell borders of neighboring cells clearly showed that OsRIP1-eGFP fusion proteins are not present in the cell wall (Figure 7, panels 1D and 2D). In contrast, the SP-eGFP construct gave rise to an intermittent GFP-signal near the cell borders. The cross-sectional fluorescent intensity plot (Figure 7, panel 3D) showed that these GFP dots coincide with the cell wall, suggesting secretion of the fusion protein. Since different cell compartments can be distinguished more easily in plant suspension cells the OsRIP1-eGFP and SP-eGFP fusion constructs were also stably transformed into *Arabidopsis* PSB-D suspension cells. While the SP-eGFP fusion proteins ended up in the vacuole, the OsRIP1-eGFP fusion protein clearly resided in the nucleocytoplasmic compartment (Appendix A).

### 2.3. Recombinant OsRIP1 Inhibits Protein Translation and Shows Selectivity Towards Non-Plant Ribosomes

Heterologous expression of OsRIP1 in *Escherichia coli* allowed the purification of recombinant His6-tagged OsRIP1. SDS-PAGE (Figure 8) revealed that the recombinant protein was highly pure and migrated with a molecular mass of approximately 31.9 kDa, which is in good agreement with the molecular mass of 31.4 kDa, calculated from the OsRIP1-HIS6 protein sequence. Western blot analysis confirmed that the polypeptide band for recombinant OsRIP1 reacted with the monoclonal antibody directed against the His tag (Figure 8). The yield of recombinant OsRIP1 amounted to approximately 1 mg per liter culture.

The purified recombinant protein was used to assess the catalytic activity of OsRIP1. To study the ability of OsRIP1 to inhibit protein translation by interacting with animal or cereal ribosomes, cell-free translation systems based on rabbit reticulocyte lysate or wheat germ extract were used, respectively. BSA was used as a negative control and had no effect on protein translation. OsRIP1 effectively inhibited protein translation in a system based on animal ribosomes in a dose dependent way, with an IC50 of 116 nM. In contrast, the translational inhibitory activity of OsRIP1 in the system based on wheat germ extract was more then 10-fold less efficient, with an IC50 estimated to be 1.5 µM. This observation suggests that cereal ribosomes are less susceptible to OsRIP1 than their animal counterparts.

## 3. Discussion

More than 30 RIP genes have been identified in the rice genome and the expression of several of them is suggested to be stress related [28,33]. Moreover, the majority of these RIPs (26 out of 38 RIP sequences identified in *Oryza sativa* ssp. japonica) are synthesized without a signal peptide suggesting they reside in the cytosol where they can possibly inactivate the plant ribosomes, raising some questions concerning their mode of action [28]. However, so far none of these RIPs has been characterized at protein level. In this work we focused on OsRIP1, a stress inducible RIP. We were able to prove that the protein resides in the cytoplasmic compartment and possesses enzymatic activity.

The residues that make up the specificity pocket of RIPs are absolutely invariant, including two tyrosine residues important for the stacking of the substrate adenine, a glutamic acid and an arginine that are the key residues for catalysis, and a tryptophan [17,34,35,36]. *In silico* analyses of the protein sequence and modeling of the protein structure suggested that OsRIP1 shows a typical RIP-fold and possesses the required catalytic residues for enzymatic activity. Docking experiments with AMP further confirmed that the active site cavity of OsRIP1 meets the requirements for enzymatic activity. The three-dimensional conformation of OsRIP1 also closely resembles that of other cereal RIPs such as the barley bRIP1 and tritin from wheat. Similar to other cereal RIPs OsRIP1 seems to lack an α-helix in the N-terminal subdomain as well as two β-sheets in the C-terminal subdomain that are present in non-cereal RIPs [22]. Furthermore, OsRIP1 possesses an extra C-terminal α-helix as reported for tritin and bRIP1 but absent in the maize RIPs. To date, the possible function of this C-terminal helix is not known. 

Despite these similarities, OsRIP1 also shows some small but potentially important differences with other cereal RIPs. At present only few cereal RIPs have been characterized in detail, but the activity of several RIPs is known to be dependent on some unique activation mechanisms relating to certain structural features. For example, the maize RIPs and the JIP60 from barley require the removal of internal peptides to be activated. For maize RIP1 it was shown that the internal peptide prevented the interaction of the RIP with the ribosomal proteins which is required for enzymatic activity. The barley bRIP1 does not have an internal peptide, but has a unique loop structure called the ‘switch region’ at a location corresponding to the inactivation region in the maize RIPs. This loop has a different conformation depending on the liganded state [30]. Although not previously described, our sequence and modeling studies suggested that tritin from wheat also harbors this same loop in its structure. Sequence and structural analyses for OsRIP1 did not confirm the presence of a similar loop or inactivation region, at least not at a similar location in the protein. However, OsRIP1 contains an extra amino acid stretch near its C-terminus that was not reported in other cereal RIPs. It cannot be excluded that this region plays a significant role in protein function, for example in the interaction with ribosomal proteins. Thus far the interaction of RIPs with ribosomal proteins has been studied only for a handful of proteins, among which only one cereal RIP (maize RIP1). The mechanism by which this interaction takes place and the domains of the RIP involved are variable [22], making it hard to elaborate on the functional relevance of this extra amino acid stretch present in OsRIP1. 

*In silico* screening of the OsRIP1 sequence using the signalP server suggested the absence of a signal peptide. While servers LOCALIZER and ESLpred predicted the protein to be localized in the chloroplasts or mitochondria, respectively, other tools such as TargetP, locTREE3 and Plant-mPLoc predicted the protein to be cytoplasmic. Using eGFP-fusion proteins we experimentally proved that OsRIP1 resides in the cytoplasm and the nucleus of *N. benthamiana* epidermal cells. The OsRIP1-eGFP fusion protein co-localized with DAPI (nuclear marker), but not with propidium iodide (cell wall marker), and showed a similar localization pattern as the free eGFP control, a protein that is often used as a nucleocytoplasmic marker [37,38]. In contrast, when free eGFP was fused with the signal peptide of a rice RIP that is predicted to be secreted (MSU LOC_Os03g48200.1), this SP-eGFP fusion protein did co-localize with propidium iodide but not with DAPI, suggesting secretion of the protein to the apoplast. Localization studies using *A. thaliana* cell cultures also showed a clear nucleocytoplasmic expression pattern for the OsRIP1-eGFP fusion while the SP-eGFP ended up in the vacuole. Taking into account that the OsRIP1 sequence does not have a classical NLS and the molecular mass of the fusion protein is about 58 kDa (31 kDa for OsRIP1 domain and 27 kDa for eGFP domain) OsRIP1-eGFP most likely does not require active transport to get in the nuclear compartment but can freely diffuse to the nucleus [39].

Since *in silico* modeling cannot provide definite proof that OsRIP1 is indeed a functional RIP, recombinant protein was produced using *E. coli* as an expression system. *E. coli* is a heterologous expression system that is frequently used to make recombinant protein. However, devoid of extensive post-translational modification and glycosylation mechanisms, this expression system is usually not preferred for more complex eukaryotic proteins. However, since eGFP-fusion proteins indicated that OsRIP1 is a nucleocytoplasmic protein it is not expected to undergo significant post-translational modifications or glycosylation, making *E. coli* a suitable expression system. To ensure that the eukaryotic codon usage was no problem during protein expression, the *E. coli* DE3 Rosetta strain containing an additional plasmid to enrich the available eukaryotic tRNA pool was preferred. In the past the *E. coli* expression system has already been used extensively to produce active RIPs. Most of these proteins were classical type-1 RIPs that are naturally secreted proteins, thus forming a bigger challenge with regard to post-translational modifications compared to the nucleocytoplasmic OsRIP1 [40,41,42,43,44]. Although in some cases RIP production in *E. coli* was compromised by toxic effects of the RIPs towards the prokaryotic ribosomes [45] this was not the case for OsRIP1, as evidenced by the fact that the cultures continued to grow well after OsRIP1 production was induced by IPTG. However, the yield of recombinant protein was only moderate, amounting to approximately 1 mg per liter culture.

Purified recombinant OsRIP1 was used to assess the protein inhibitory activity on translation. The effect of increasing amounts of OsRIP1 on the in vitro translation of luciferase was quantified with a luminometer. A dose dependent inhibitory effect on translation was measured and the IC50 was determined to be 116 nM. Typical IC50 values for RIPs in the same rabbit reticulocyte lysate system range from picomolar to micromolar range. More active RIPs have an IC50 in the lower nM range, including cereal RIPs such as tritin, barley, and maize. However, several RIPs have been described with an IC50 comparable or higher to OsRIP1. Zhang et al. [46] reported the IC50 of Foetidissimin II, a type-2 RIP from Cucurbitaceae, to be 252 nM. E-momorcharin and F-momorcharin have IC50s of 170 nM and 55 nM, respectively [47]. The CIP-34 from *Clerodendrum inerme* has an IC50 of 87 nM [48] and the IC50s of the type-1 and type-2 RIPs from *Malus domestica* were determined to be 175 nM and 263 nM, respectively [49]. The rather moderate activity of OsRIP1 can indicate that the extra amino acid stretch found in the OsRIP1 protein sequence is indeed an inactivation region. However, this is merely a hypothesis and it should be confirmed experimentally that the RIP is indeed more active after removal of this region.

The localization of OsRIP1 in the cytoplasmic compartment implies that the RIP can make contact with ribosomes, and raises some questions about the biological activity of the protein. Therefore, the translational inhibitory activity of the recombinant protein towards wheat ribosomes was checked using a cell free system based on wheat germ extract. The activity of OsRIP1 on wheat ribosomes was much weaker compared to rabbit ribosomes, suggesting that wheat ribosomes are not a preferred substrate for OsRIP1. Similarly, both the pro-form as well as the activated form of the maize RIP1 are not very active on conspecific ribosomes [23]. This also suggests that the removal of the inactivation loop is not a measure to protect the host ribosomes against the RIP [44]. At present, no results are available for the effect of the second maize RIP on conspecific ribosomes. The ribosomes of barley were shown to be tolerant to the effect of the barley seed RIPs [20]. Also, tritin-S found in wheat seeds is not active on conspecific ribosomes, while the leaf form did show depurination of plant rRNAs [24].

Although these findings suggest that OsRIP1 probably exerts its function by acting on ribosomes of other organisms rather than those from the rice plants itself, there is also evidence that the activity of RIPs on host ribosomes can be triggered by stress. Indeed, the unprocessed form of JIP60 only acts on barley ribosomes as a dissociation factor under certain conditions. Ribosomes from leaves exposed to methyl jasmonate or osmotically stressed or desiccated leaves are cleaved by JIP60, while ribosomes from unstressed leaves are not. In this way JIP60 functions as a molecular switch to alter the translational machinery under stress conditions [26]. At this point it cannot be excluded that OsRIP1 has a similar mode of action, and under certain conditions can act on conspecific ribosomes. It is documented that a lot of different genes encoding different forms of ribosomal proteins are found in the rice genome. Furthermore, expression of these genes was reported to be stress related [50]. Given the fact that the specificity of RIPs depends on their ability to interact with these ribosomal proteins and that this interaction is quite complex, it is possible that OsRIP1 can interact with rice ribosomes under certain circumstances depending on their ribosomal protein composition.

## 4. Materials and Methods 

### 4.1. Plant Materials

*Nicotiana benthamiana* seeds were kindly provided by dr. Verne A. Sisson (Oxford Tobacco Research station, Oxford, NC, USA). The seeds were germinated in a pot containing wet commercial soil in a growth chamber at 28 °C with a 16/8 h light/dark photoperiod. Two weeks after germination plants were transferred to individual pots and grown in the growth chamber. *Arabidopsis thaliana* PSB-D suspension cells ecotype Landsberg erecta were obtained from the department of Plant Systems Biology (Vlaams Instituut voor Biotechnologie, Zwijnaarde, Belgium). The suspension culture was maintained on a 7-day culture cycle by adding 5 mL of the cell culture to 45 mL of medium containing 4.43 g/L Murashige-Skoog Basal salts with minimal organics (Sigma-Aldrich, Saint Louis, MO, USA), 30 g/L sucrose, 0.5 mg/L α-naphthalene acetic acid, and 0.05 mg/L kinetin, at pH 5.7. The cells were grown on a rotary shaker (150 rpm) at 25 °C in the dark.

### 4.2. Construction of Expression Vectors

The OsRIP1 coding sequence (LOC_Os01g06740) was amplified using genomic DNA extracted from wild-type *Oryza sativa*, spp. japonica cv Nipponbare plants by nested PCR with Q5 polymerase (New England Biolabs, Ipswich, MA, USA). Primers P43/P42 and P43/ P44 (Appendix A) were used in the first and second PCR respectively. Cycling parameters were 98 °C—30 s; 30× (98 °C—10 s; 65 °C—30 s; 72 °C—30 s); and 72 °C—2 min. PCR products were ligated in the pJET1.2 vector using the CloneJET PCR Cloning Kit (Thermo Fisher Scientific, Waltham, MA, USA). *E. coli* TOP10 cells were transformed with the recombinant vectors using heat-shock transformation. Transformants were screened by standard colony PCR using primers EVD275/EVD276 (Appendix A). Plasmids were purified with the GeneJET Plasmid Miniprep kit (Thermo Fisher Scientific) and sequenced (LGC Genomics, Berlin, Germany). 

### 4.3. Localization Constructs

Vectors for expression of OsRIP1 N-and C-terminal eGFP fusion proteins under the control of the CaMV 35S promoter were constructed using Gateway Cloning Technology (Thermo Fisher Scientific). Two consecutive PCRs were performed to attach attB sites to the OsRIP1 coding sequence. In the first PCR, the sequence of OSRIP1 was amplified using Q5 polymerase (New England Biolabs) and primers without stop codon (P355/P356, Appendix A) or with stop codon (P355/P357, Appendix A). Afterwards, primers EVD2 and EVD4 (Appendix A) were used in the second PCR to complete the attB sites. The sequence of the signal peptide of the secreted RIP (LOC_Os03g48200.1) containing partial attB sites, was ordered as an oligo DNA molecule (Sigma-Aldrich) and attB sites were completed by PCR using primers EVD2 and EVD4. Cycling parameters for all PCRs were: 98 °C—30 s; 30× (98 °C—10 s; 65 °C—30 s; 72 °C—30 s); and 72 °C—2 min. The resulting PCR products were cloned into the pDONR221 vector in a BP recombination reaction by incubating equimolar amounts of attB PCR products and donor vector overnight together with BP clonase II enzyme mix (Thermo Fisher Scientific). Entry clones were transformed into heat-shock competent *E. coli* TOP10 cells and transformants were screened by standard colony PCR using primers EVD386-EVD387 (Appendix A). Plasmids were purified with the GeneJET Plasmid Miniprep kit (Thermo Fisher Scientific) and sequenced (LGC genomics). Entry clones containing OsRIP1 constructs without stop codon and entry clones containing the SP-construct were recombined with the pK7FWG2.0 destination vector to obtain expression clones for C-terminal eGFP fusion proteins. OSRIP1 entry clones with stop codon were recombined with the pK7WGF2.0 destination vector to obtain expression clones for N-terminal eGFP fusion proteins. Expression clones were transformed into heat-shock competent *E. coli* TOP10 cells and transformants were screened by standard colony PCR using primers EVD472-P1 (Appendix A). A binary vector expressing free eGFP under control of the CaMV 35S promoter (pK7WG2 vector) was used as a control in the localization studies.

### 4.4. Construct for Recombinant Protein Production

The OsRIP1 coding sequence ligated in the pJET1.2 vector (Thermo Fisher Scientific) was used as a template to prepare the OsRIP1 His6-tagged expression construct. DNA sequences encoding a C-terminal Gly3-linker followed by a 6xHis-tag, as well as Gibson assembly sites allowing recombination into the pET21a vector by Gibson assembly, were added to the ORF by two consecutive PCRs. The used primers for the first and second PCR were P359/P358 and P359/P216 (Appendix A), respectively. Cycling parameters for the PCR reactions were 98 °C—30 s; 35× (98 °C—10 s; 68 °C—30 s; 72 °C—30 s); 72 °C—2 min and Q5 (New England Biolabs) was used as DNA polymerase. Next, PCR products were purified using the Analytik Jena purification kit (Analytik Jena, Jena, Germany) and the DNA concentration was measured with a Nanodrop ND- 1000 (Thermo Fisher Scientific) at 260 nm. The pET21a vector backbone was linearized by PCR using the primers P242/P243 (Appendix A) using Q5 polymerase and following cycling parameters: 98 °C—30 s; 35× (98 °C—10 s; 68 °C—30 s; 72 °C—3 min); 72 °C—5 min. The PCR product was purified by gel extraction (QIA-quick kit, Qiagen, Hilden, Germany). The OsRIP1-His6 sequence was subsequently ligated in the pET21a vector using a Gibson assembly reaction (New England Biolabs). After the reaction, half of the Gibson assembly mixture was transformed into heat shock competent cells of *E. coli* strain Rosetta (DE3) (Novagen, Merck (MSD), Kenilworth, NJ, USA).

### 4.5. Transient Transformation of N.benthamiana

Expression clones containing the different eGFP fusion constructs, as well as the free eGFP control, were introduced into *Agrobacterium* strain LBA4404 using tri-parental mating. *A. tumefaciens* cultures harboring the different expression vectors were grown overnight in 5 mL liquid YEB medium with 50 µg/mL spectinomycin on a rotary shaker (200 rpm) at 26 °C. The next day this culture was used as a preculture to inoculate 20 mL of Yeast Extract Broth (YEB) containing 50 µg/mL spectinomycin. The culture was grown until an OD600 of 0.8 was reached and then 2 mL of bacterial culture was harvested by centrifugation (7000 rpm, 10 min). Cells were resuspended in infiltration medium (10 mM MES, 2 mM Na_2_HPO_4_, 0.5% (*m*/*v*) glucose, pH 5.6) and centrifuged again (7000 rpm, 10 min). This washing step was repeated twice. Washed cells were diluted in infiltration medium supplemented with 100 µM acetosyringone to reach an OD600 of 0.4–0.5. The resuspended *Agrobacteria* were kept at room temperature for 1–2 h before infiltration. The *Agrobacterium* suspension was successively used for the infiltration of the abaxial epidermis of young leaves of 4–6-week-old *N. benthamiana* plants. Two days post-infiltration, plants infiltrated with *Agrobacteria* harboring eGFP fusion constructs were either directly used for microscopic analysis or were co-infiltrated with DAPI (10 µg/mL in H_2_O) or propidium iodide (10 µg/mL in H_2_O) and then used in microscopic studies.

### 4.6. Stable Transformation of A. thaliana PSB-D Cells

*Agrobacterium* LBA4404 containing the different OsRIP1 fusion constructs was used for transformation of *A. thaliana* suspension cells ecotype Landsberg erecta by a callus-free co-cultivation method [51]. 

### 4.7. OsRIP1 Protein Production

Recombinant *E. coli* Rosetta cells containing the OsRIP1-HIS6 constructs in the pET21a vector were grown overnight in 5 mL LB supplemented with 25 µg/mL carbenicillin and 25 µg/mL chloramphenicol at 37 °C on a rotary shaker at 185 rpm. The next morning, the *E. coli* pre-cultures were used to inoculate cultures of 300 mL LB supplemented with 25 µg/mL carbenicillin and 25 µg/mL chloramphenicol and these cultures were grown at 37 °C on a rotary shaker (185 rpm). When an OD600nm of 0.6–0.8 was reached, recombinant protein expression was induced with 0.2 mM isopropyl β-D-1-thiogalactopyranoside (IPTG) (MP Biomedicals, Valiant, Yantai, Shandong, China). After induction, cultures were grown for 72 h at 14 °C on a rotary shaker at 185 rpm. Subsequently cells were harvested by centrifugation at 8000 rpm for 20 min and the cell pellets were pooled and kept overnight at −20 °C. The next day the cell pellet was thawed and resuspended in phosphate buffer (PB; 0.02 M NaH_2_PO_4_∙2H_2_O, 0.23 M Na_2_HPO_4_) at pH 8 containing 1 mg/mL lysozyme to facilitate cell lysis. The solution was sonicated and afterwards centrifuged at 4 °C for 45 min to remove cell debris and the insoluble protein fraction. The soluble fraction was used for recombinant OsRIP1 protein purification.

### 4.8. OsRIP1 Protein Purification

OsRIP1 was purified by two consecutive immobilized metal affinity chromatography steps. First the soluble protein solution was loaded on a Ni-NTA (MCLAB, South San Francisco, California) agarose column (Ø 28 mm, height 15 mm) equilibrated with PB at pH 8. Histidine tags have a high affinity for nickel ions, as a result the His6-tagged recombinant protein will be retained on the column. The column was washed with PB, 50 mM imidazole, pH 8 to remove aspecific proteins and elution of the column was performed with PB, 250 mM imidazole pH 8. About 250 mL of eluate was dialyzed overnight against PB with 0.5 M NaCl, pH 8. This purification step was repeated using a smaller Ni-NTA column (Ø 18 mm, height 15 mm) and PB buffers containing 0.5M NaCl. Elution fractions of 1–2 mL were collected and the purity of the purified protein was assessed by SDS-PAGE and western blot analysis. For protein synthesis inhibition assays, the elution fraction with the highest protein concentration was used for desalting and buffer exchange by using DPBS buffer (Sigma-Aldrich) and the Zeba spin columns (Thermo Fisher Scientific).

### 4.9. SDS-PAGE and Western Blot

Protein samples were analyzed by SDS-PAGE on 15% acrylamide gels as described by Laemmli (1970). After separation, proteins were visualized by gel staining with Coomassie Brilliant Blue R-250 or blotted onto polyvinylidene fluoride transfer membranes (FluoroTrans^®^ PVDF, Pall Laboratory, USA). Membranes were blocked with Tris-buffered saline (TBS; 10 mM Tris, 150 mM NaCl, 0.1% (*v*/*v*) Triton X-100, pH 7.6) containing 5% (*w*/*v*) non-fat milk powder. Subsequently, membranes were incubated for 1 h with a mouse monoclonal anti-His6 antibody (Thermo Fisher Scientific) diluted 1/3000 in TBS. After washing three times with TBS, membranes were incubated for 1 h with the 1/1000 diluted rabbit anti-mouse IgG secondary antibody labelled with horseradish peroxidase (Dako, Glostrup, Denmark). After washing two times with TBS and one time with 0.1 M Tris-HCl buffer (pH 7.6), immunodetection was achieved using a colorimetric assay with 0.1 M Tris-HCl buffer (pH 7.6) containing 700 µM 3,3′-diaminobenzidine tetrahydrochloride (Sigma-Aldrich) and 0.03% (*v*/*v*) hydrogen peroxide. The detection reaction was stopped after 2–10 min by rinsing the membrane with distilled water. All washes and incubations were performed on a gently shaking platform at room temperature.

### 4.10. Protein Synthesis Inhibition Assay

The protein synthesis inhibition activity of the recombinant OsRIP1 and controls (DPBS buffer and BSA) was determined using an in vitro transcription/translation system [49]. To test the effect of OsRIP1 on animal ribosomes the TnT^®^ T7 Quick Coupled Transcription/Translation System Kit (Promega, Mannheim, Germany) based 

On a cell-free system derived from rabbit reticulocytes. The effect on plant ribosomes was analyzed using the TnT^®^ T7 Coupled Wheat Germ Extract kit (Promega) was used. According to the manufacturer’s instructions, the prepared mixture was incubated at 30°C for 10 min and chilled on ice. Afterwards, 2 µL DPBS buffer or buffer containing different concentrations of OsRIP1 was added to the reaction mixture and incubated for 30 min at 30 °C. After addition of 35 µL nuclease-free water at room temperature, the reaction samples were transferred to a luminometer plate (Greiner Labortechnik, Frickenhausen, Germany) containing 5 µL luciferase assay reagent at 25 °C. The relative luciferase activities of the samples were determined using a microtiter top plate reader (Infinite F200 Pro, Tecan, Mannedorf, Switzerland).

### 4.11. Confocal Microscopy and Image Analysis

Images were acquired with a Nikon A1R confocal laser scanning microscope (Nikon Instruments) mounted on a Nikon Ti-E inverted epifluorescence body with a 40 × S Plan Fluor ELWD air objective lens (NA 0.60) or a 60× Plan APO VC water immersion lens (NA 1.20). eGFP was excited with a 488 nm argon ion laser, propidium iodide was excited with a 560 nm laser and DAPI was excited with a 404 nm laser. Emission filters were 500–550 nm for eGFP, 570–620 nm for propidium iodide and 425–475 nm for DAPI. Image analysis and assessment of co-localization were performed in Fiji [52].

### 4.12. In Silico Analyses

The presence of a signal peptide was predicted by means of the SignalP server [29]. TargetP 1.1, LocTREE3, Plant-mPLoc, LOCALIZER, and ESLpred servers were used to predict protein subcellular localization [53,54,55,56,57]. Multiple sequence alignment was performed using the Clustal Omega program [58]. Three dimensional models of different RIPs were predicted based on their coding sequence using the Phyre2 web portal for protein modeling, prediction and analysis [31]. Homology modeling of OsRIP1 from *Oryza sativa* was performed with the YASARA Structure program [59], using the barley bRIP-I (PDB codes 4FBC and 4FB9), as a template [30]. Five different models were built from each of the templates and, finally, a single hybrid model of OsRIP1 was built up from the ten previous models. PROCHECK [60], ANOLEA [61], and the calculated QMEAN score [62,63], were used to assess the geometric and thermodynamic qualities of the OSRIP1 model. In this respect, (1) a single amino acid residue (A242) over 282 occurred out of the generously-allowed region in the Ramachandran plot, (2) twenty five amino acid residues over 282 exhibited an energy over the threshold value and, and (3) an acceptable QMEAN score of −1.45 was calculated for the OsRIP1 model. Docking of AMP to the modeled OsRIP1 was performed with the YASARA structure program. docking experiments were performed at the SwissDock web server (http://www.swissdock.ch)) [64,65]. Molecular cartoons were drawn with Chimera [66].

## Figures and Tables

**Figure 1 toxins-11-00325-f001:**
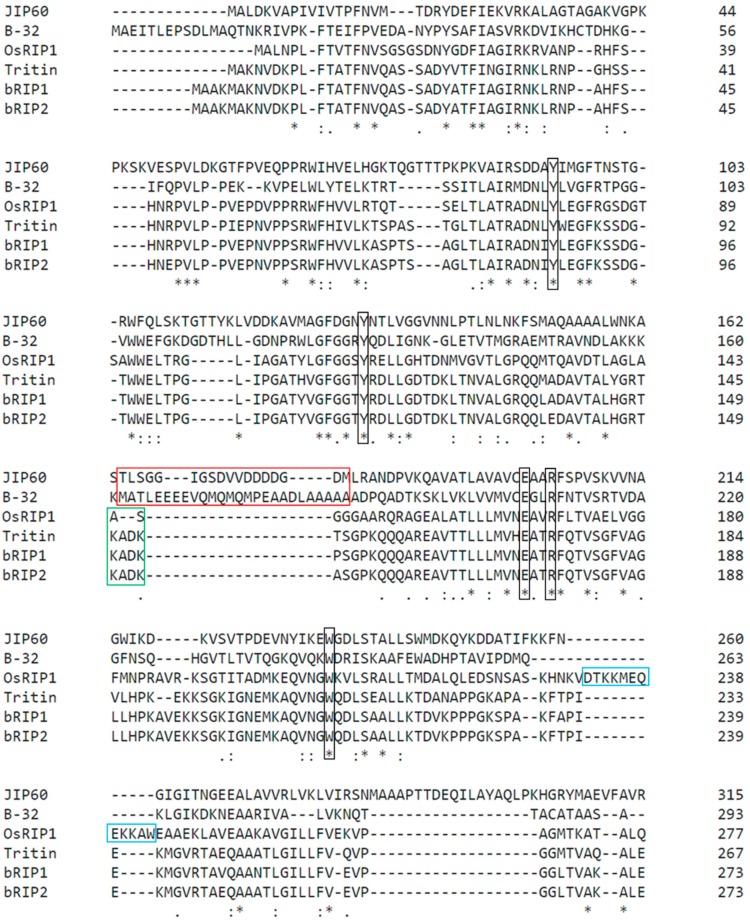
Multiple sequence alignment of the coding sequences for OsRIP1 and other cereal ribosome-inactivating proteins (RIPs): JIP60 from barley (GenBank no. X66376), B-32 or maize RIP1 (GenBank no. M83926) tritin from wheat (GenBank no. D13795), and the barley seed RIPs barley RIP1 (bRIP1) (GenBank no. M62905) and bRIP2 (GenBank no. AAA32951.1). Since JIP60 is a type-3 RIP only the RIP domain is represented (first 375 amino acids). Black boxes highlight the invariant active site residues described for RIPs. The red box shows the known inactivation regions that are proteolytically removed in JIP60 and maize RIP1 (B-32). The green box shows the region that was described as a “switch” region in the barley seed RIPs. The blue box shows an extra stretch of amino acids that is present in OsRIP1 but not in other cereal RIPs.

**Figure 2 toxins-11-00325-f002:**
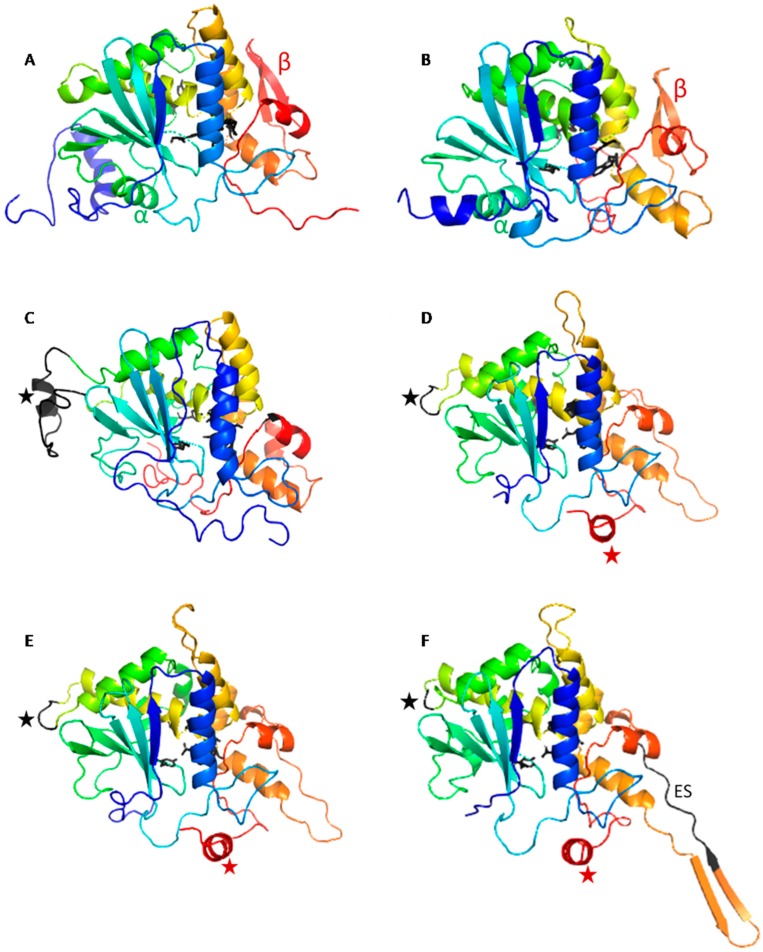
3D models for ricin A-chain (RTA) (**A**, PDB code 1RTC), trichosanthin (TCS) (**B**, PDB code 1TCS), maize RIP1 (**C**, PDB code 2PQI), Tritin (**D**, based on GenBank no. D13795), bRIP1 (**E**, PDB code 4FBC), and OsRIP1 (**F**, based on coding sequence: LOC_Os01g06740) were predicted based on their coding sequence using the Phyre2 web portal for protein modeling, prediction, and analysis (Kelley et al., 2015). Rainbow coloring was used to highlight the orientation of the structure (N-terminus = blue, C-terminus = red). The active site residues are represented in black. The α-helix and β-sheets present in non-cereal RIPs but typically absent in cereal RIP structures are indicated in the ricin-A chain and TCS structures. The inactivation region in maize RIP1, the switch region in bRIP1 and the corresponding location in OsRIP1 are indicated with a black star. The extra C-terminal helix present in bRIP1, tritin, and OsRIP1 is highlighted with a red star. The extra amino acid stretch (ES) present in OsRIP1 is highlighted in black.

**Figure 3 toxins-11-00325-f003:**
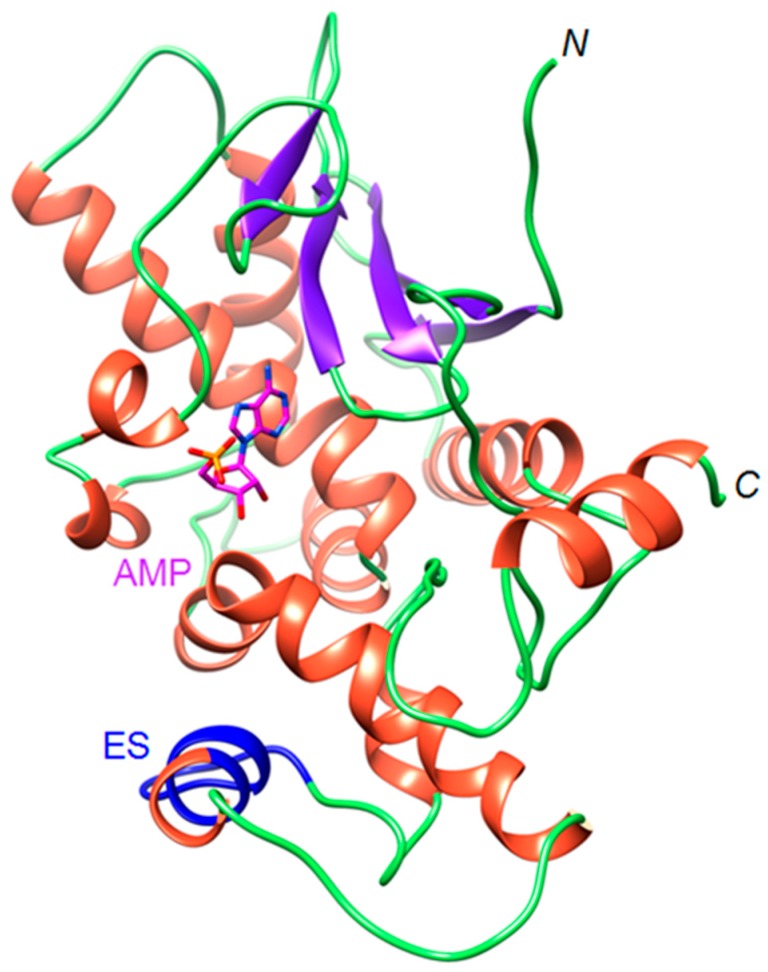
Ribbon diagram of the modeled OsRIP1 in complex with adenosine monophosphate (AMP), showing the localization of the extra sequence ES (colored blue) occurring along the polypeptide chain of OsRIP1. N and C correspond to the N-terminal and C-terminal ends of OsRIP1, respectively.

**Figure 4 toxins-11-00325-f004:**
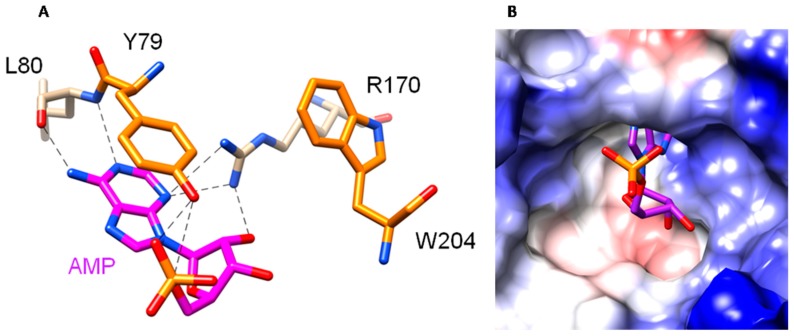
Molecular docking of AMP. (**A**). Docking of AMP to the active site of OsRIP1, showing the network of H-bonds (dashed lines) anchoring AMP to OsRIP1. Aromatic residues Y79 and R170 involved in stacking interactions with AMP are colored orange. (**B**). Coulombic surface coloring of the molecular surface surrounding the active site cavity of OSRIP1.

**Figure 5 toxins-11-00325-f005:**
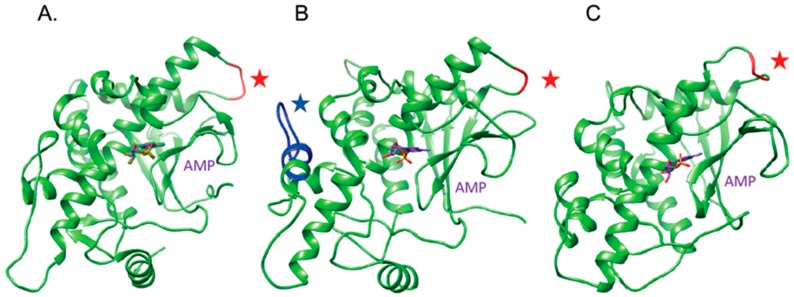
Molecular modeling of cereal RIP sequences and docking with AMP. (**A**). Ribbon diagram of the barley RIP1 (PDB code 4FBC) in complex with AMP (colored purple). The switch region 150–153 is colored red and indicated by a red star. (**B**). Ribbon diagram of the modeled OsRIP1 in complex with AMP (colored purple). The shorter switch region 144–145 is colored red and indicated by a red star. The extra-sequence 232–243 occurring in the C-terminal part of OsRIP1 is colored in blue and indicated by a blue star. (**C**). Ribbon diagram of maize RIP1 (PDB code 2PQI) in complex with AMP (colored purple). The region that remains from the proteolytic removal of the 162–187 amino acid sequence stretch is colored red and indicated by a red star.

**Figure 6 toxins-11-00325-f006:**
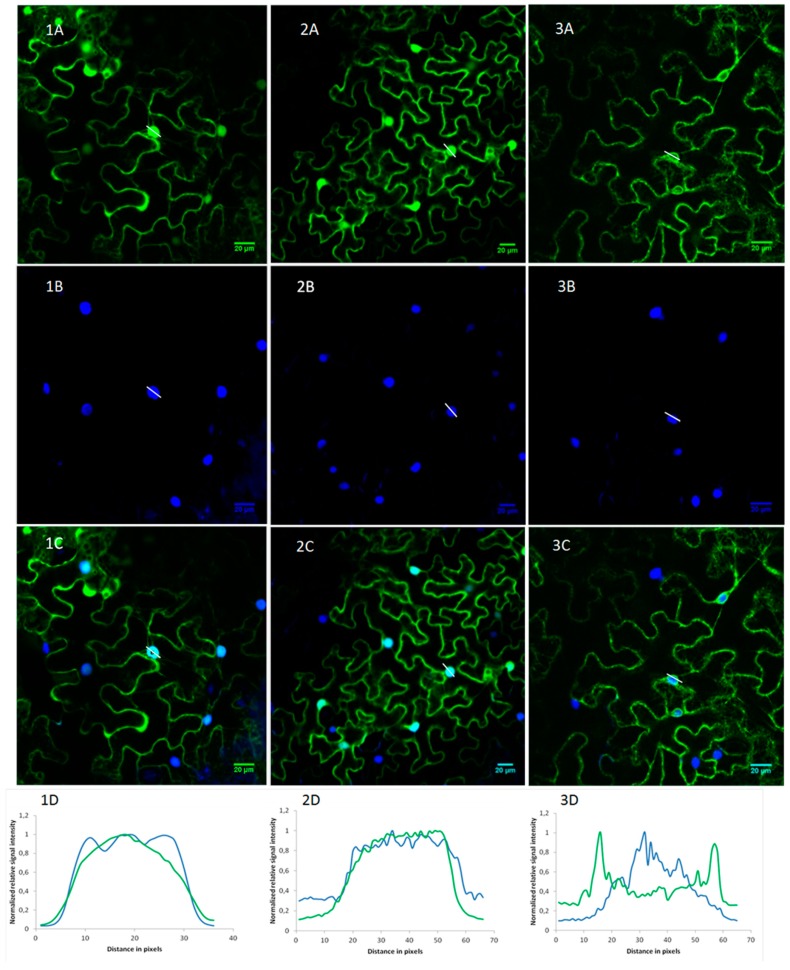
Subcellular localization of free eGFP (**1A**), OsRIP1-eGFP (**2A**), and SP-eGFP (**3A**) in *N. benthamiana* epidermal cells. Nuclei were stained with DAPI (**1B**–**3B**) and co-localization of DAPI and eGFP fusion proteins was assessed in overlay pictures (**1C**–**3C**). Panels (**1D**–**3D**) show the plot profiles of the eGFP and DAPI fluorescent signals across the nuclei.

**Figure 7 toxins-11-00325-f007:**
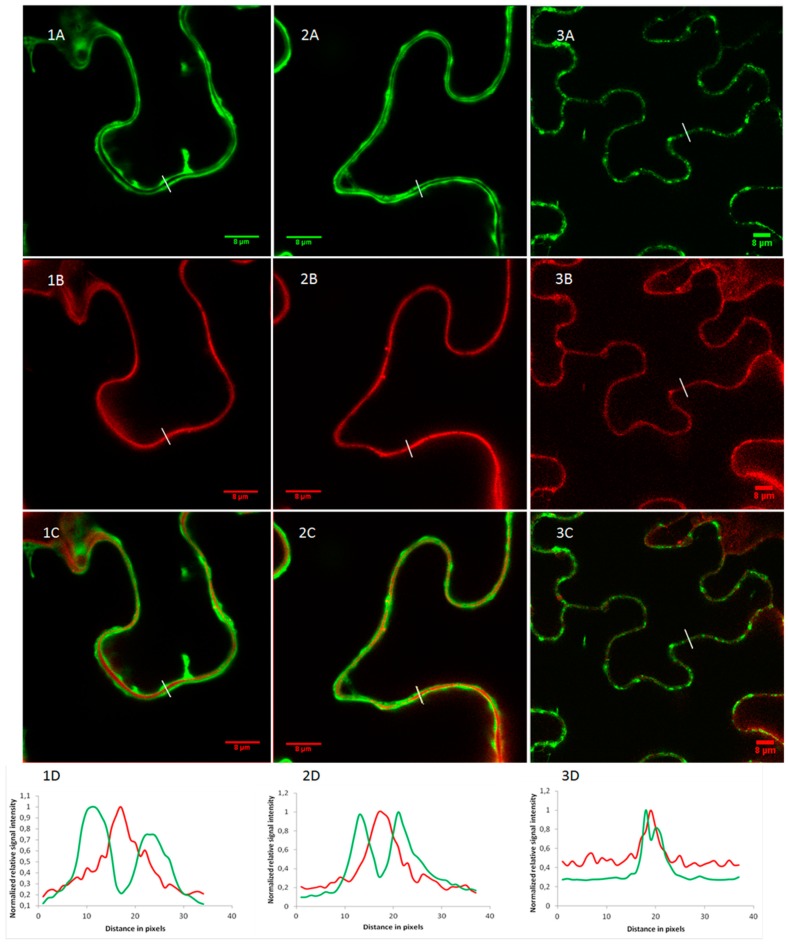
Subcellular localization of OsRIP1-eGFP (**1A**,**2A**) and SP-eGFP (**3A**) in *N. benthamiana* epidermal cells. Cell walls were stained with propidium iodide (**1B**–**3B**) and co-localization of propidium iodide and eGFP fusion proteins was assessed in overlay pictures (**1C**–**3C**). Panels (**1D**–**3D**) show the plot profiles of the eGFP and propidium iodide fluorescent signals at the cell borders.

**Figure 8 toxins-11-00325-f008:**
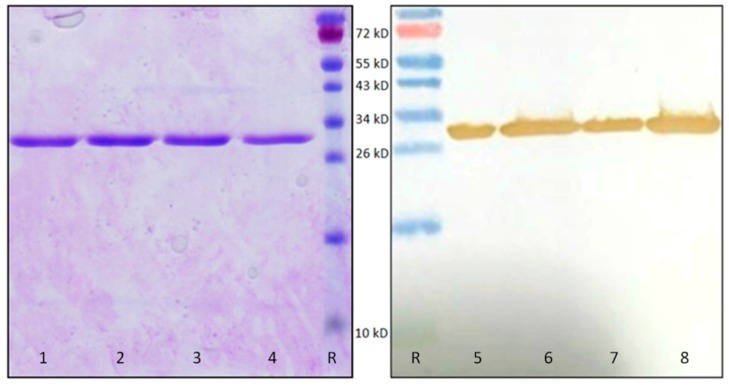
SDS-PAGE (lanes **1**–**4**) and western blot (lanes **5**–**8**) analysis of purified recombinant OsRIP1. Four elution fractions containing about 4 µg of recombinant OsRIP1 were analyzed after the second affinity chromatography. A prestained protein marker (lanes R) (Thermo Fisher Scientific) was used as a reference. The western blot analysis was performed using an anti-HIS antibody (Thermo Fisher Scientific).

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
