# Peer review of "Structure and Activity of a Cytosolic Ribosome-Inactivating Protein from Rice"

_toxins, 2019, doi:10.3390/toxins11060325_

Round 1
Reviewer 1 Report
In the manuscript entitled “Structure and Activity of a Cytosolic Ribosome-Inactivating Protein from Rice ” the authors describe for the first time the RIP from rice both the protein structure and the biological activity.
In the introduction the authors describe the toxic effect of plants RIPs and cite old papers; recently Giansanti et al published two detailed and updated reviews on this topic.
· Giansanti F, Flavell DJ, Angelucci F, Fabbrini MS, Ippoliti R. Strategies to Improve the Clinical Utility of Saporin-Based Targeted Toxins. Toxins (Basel). 2018 Feb 13;10(2).
· Giansanti F., Di Leandro L., Koutris I., Cialfi A., Benedetti E., Laurenti G., Pitari G., Ippoliti R. “Ricin and Saporin: Plant Enzymes for the Research and the Clinics”. Current Chemical Biology, 2010 May; 4(2): 99-107. (doi: 10.2174/187231310791170801).
From the experimental point of view I ask the authors if they have done toxicity tests using prokaryotic ribosomes (se figure 9) and what was the yield of OsRIP1–HIS6 (mg/lt of culture medium) in E. coli DE3 Rosetta strain (Paragraph 2.2).
To check the purity of OsRIP1–HIS6 purified from E. coli DE3 Rosetta strain the authors perform a classical SDS-PAGE and a western blotting using as revealing system Diaminobenzidine (see 10, figure 8 and page 16 materials an methods). Why did the authors not use more sensitive techniques for blotting revelation, as ECL, to check the purity of the protein and the absence of aggregates and / or degradation products?
For these reasons, after the corrections/additions requested the manuscript can be accepted.
Minor revisions:
Page 7, lane 178 please Nicotiana benthamiana in italics
Page 10, lane 213 please write Escherichia coli in italics
Page 16, lane 477 please write in vitro in italics
Author Response
In the manuscript entitled “Structure and Activity of a Cytosolic Ribosome-Inactivating Protein from Rice ” the authors describe for the first time the RIP from rice both the protein structure and the biological activity.
In the introduction the authors describe the toxic effect of plants RIPs and cite old papers; recently Giansanti et al published two detailed and updated reviews on this topic.
·
Giansanti F, Flavell DJ, Angelucci F, Fabbrini MS, Ippoliti R. Strategies to Improve the Clinical Utility of Saporin-Based Targeted Toxins. Toxins (Basel). 2018 Feb 13;10(2).
·
Giansanti F., Di Leandro L., Koutris I., Cialfi A., Benedetti E., Laurenti G., Pitari G., Ippoliti R. “Ricin and Saporin: Plant Enzymes for the Research and the Clinics”. Current Chemical Biology, 2010 May; 4(2): 99-107. (doi: 10.2174/187231310791170801).
>>The citations to the older articles have been replaced by the suggested citations.
From the experimental point of view I ask the authors if they have done toxicity tests using prokaryotic ribosomes (se figure 9) and what was the yield of OsRIP1–HIS6 (mg/lt of culture medium) in E. coli DE3 Rosetta strain (Paragraph 2.2).
>>We did not test the effect of recombinant OsRIP1 towards prokaryotic ribosomes since this was not really part of our research question. The research focused on the effect of RIPs on eukaryotic, in particular plant, ribosomes. However, since the recombinant OsRIP1 is produced intracellularly in E. coli and no adverse effects on bacterial growth were observed after induction with IPTG, one can assume that OsRIP1 is not (very) active on prokaryotic ribosomes. In the discussion part, we have already elaborated on this topic. The yield of recombinant OsRIP1 (about 1mg per liter culture) was already mentioned in the discussion part, but has now also been mentioned in the results section.
To check the purity of OsRIP1–HIS6 purified from E. coli DE3 Rosetta strain the authors perform a classical SDS-PAGE and a western blotting using as revealing system Diaminobenzidine (see 10, figure 8 and page 16 materials an methods).
Why did the authors not use more sensitive techniques for blotting revelation, as ECL, to check the purity of the protein and the absence of aggregates and / or degradation products?
>>Western blot analysis (using anti-HIS antibody) was used to be certain that the purified protein was indeed the OsRIP1-HIS6. Since SDS-PAGE indicated that purified OsRIP1 was very pure (>95%), we did not check more sensitive techniques for western blot detection. We agree with the reviewer that more sensitive techniques would be advisable in case there would have been evidence for degradation or impurities.
For these reasons, after the corrections/additions requested the manuscript can be accepted.
Minor revisions: Page 7, lane 178 please Nicotiana benthamiana in italics Page 10, lane 213 please write Escherichia coli in italics Page 16, lane 477 please write in vitro in italics
>>These corrections were made in the revised manuscript
Reviewer 2 Report
In this manuscript, the authors report the cloning, expression in E. coli and purification of OsRIP1 from the genome of Oryza sativa spp. japonica. The obtained protein inhibits (poorly) protein synthesis in rabbit reticulocyte lysate and shares sequence homology with type 1 ribosome-inactivating proteins (RIPs) from cereals. They also report a 3D model for OsRIP1 obtained by comparative modelling and study the localization of OsRIP1 in transformed Nicotiana benthamiana epidermal cells and Arabidopsis thaliana suspension cells.
General comments:
Although many RIPs have been described, the information presented in this manuscript is interesting because the authors carry out a quite complete study of the structure and localization of a RIP from rice.
On this basis, the paper deserves publication with major changes.
Specific comments:
Abstract and introduction
The protein has not been obtained from rice but from a recombinant bacteria and the localization has been studied in Nicotiana benthamiana epidermal cells and Arabidopsis thaliana suspension cells. Both facts must be indicated in the abstract and introduction.
Figure 1
The quality of this figure must be improved.
Figure 8
Authors must explain the meaning of the different lanes.
Figure 9
This figure is not necessary because does not provide relevant information and the IC50 is indicated in the text.
Figures S1 and S2
The authors must clarify and indicate in the legends the meaning of the pictures of these figures.
Author Response
The protein has not been obtained from rice but from a recombinant bacteria and the localization has been studied in Nicotiana benthamiana epidermal cells and Arabidopsis thaliana suspension cells. Both facts must be indicated in the abstract and introduction. >>As requested by the reviewer these facts have now been mentioned explicitly in the abstract and the introduction of the revised manuscript. Figure 1 The quality of this figure must be improved. >>The new figure 1 was made at higher quality Figure 8 Authors must explain the meaning of the different lanes. >>An explanation of the lanes was included in the legend to the figure Figure 9 This figure is not necessary because does not provide relevant information and the IC50 is indicated in the text. >>As suggested by the reviewer, this figure was deleted. Figures S1 and S2 The authors must clarify and indicate in the legends the meaning of the pictures of these figures. >>The legends for these figures have been extended and clarified.Round 2
Reviewer 1 Report
After corrections and explanations provided by the authors, the work can be accepted